# Non-Syndromic Autosomal Dominant Hearing Loss: The First Italian Family Carrying a Mutation in the *NCOA3* Gene

**DOI:** 10.3390/genes12071043

**Published:** 2021-07-06

**Authors:** Paola Tesolin, Anna Morgan, Michela Notarangelo, Rocco Pio Ortore, Maria Pina Concas, Angelantonio Notarangelo, Giorgia Girotto

**Affiliations:** 1Department of Medicine, Surgery and Health Sciences, University of Trieste, 34127 Trieste, Italy; giorgia.girotto@burlo.trieste.it; 2Institute for Maternal and Child Health—I.R.C.C.S. “Burlo Garofolo”, 34137 Trieste, Italy; anna.morgan@burlo.trieste.it (A.M.); mariapina.concas@burlo.trieste.it (M.P.C.); 3Department of Cellular, Computational and Integrative Biology (CIBIO), University of Trento, 38123 Trento, Italy; michela.notarangelo@unitn.it; 4UOC Otolaryngology, Institute I.R.C.C.S. “Casa Sollievo della Sofferenza”, 71013 San Giovanni Rotondo, Italy; r.ortore@operapadrepio.it; 5UOC Medical Genetics, Institute I.R.C.C.S. “Casa Sollievo della Sofferenza”, 71013 San Giovanni Rotondo, Italy; a.notarangelo@operapadrepio.it

**Keywords:** non-syndromic hearing loss, *NCOA3* gene, exome sequencing, Italian family, autosomal dominant inheritance

## Abstract

Hearing loss (HL) is the most frequent sensory disorder, affecting about 1–3 per 1000 live births, with more than half of the cases attributable to genetic causes. Despite the fact that many HL causative genes have already been identified, current genetic tests fail to provide a diagnosis for about 40% of the patients, suggesting that other causes still need to be discovered. Here, we describe a four-generation Italian family affected by autosomal dominant non-syndromic hearing loss (ADNSHL), in which exome sequencing revealed a likely pathogenic variant in *NCOA3* (NM_181659.3, c.2909G>C, p.(Gly970Ala)), a gene recently described as a novel candidate for ADNSHL in a Brazilian family. A comparison between the two families highlighted a series of similarities: both the identified variants are missense, localized in exon 15 of the *NCOA3* gene and lead to a similar clinical phenotype, with non-syndromic, sensorineural, bilateral, moderate to profound hearing loss, with a variable age of onset. Our findings (i.e., the identification of the second family reported globally with HL caused by a variant in *NCOA3*) further support the involvement of *NCOA3* in the etiopathogenesis of ADNSHL, which should, thus, be considered as a new gene for autosomal dominant non-syndromic hearing loss.

## 1. Introduction

Hearing loss (HL) is the most frequent sensory disorder, affecting about 1–3 per 1000 live births [1]. In developed countries, up to 80% of congenital cases are attributable to genetic defects, which can lead to syndromic hearing loss with other organ abnormalities, or non-syndromic HL, which represents the majority of the cases (i.e., 70%, although recent findings suggest that this percentage might be lower) [2]. 

Non-syndromic hearing loss (NSHL) can be inherited as an autosomal recessive disease (i.e., 80%), autosomal dominant (i.e., 15–20%) or X-linked, mitochondrial or Y-linked, which, however, account for less than 1% of the total cases [1]. Definition of the molecular cause of NSHL is hampered by the high clinical and genetic heterogeneity of the disease, with more than 123 genes and 170 loci described thus far (Hereditary Hearing Loss Homepage, https://hereditaryhearingloss.org/, accessed on 25 May 2021). Despite technological advances, the current genetic tests fail to provide a diagnosis for about 40% of patients, suggesting that novel genes and mutations still need to be identified [3]. In particular, several studies demonstrated a lower detection rate for autosomal dominant NSHL families (ADNSHL) [4,5], which might be due to the phenotype variability among patients that leads to difficulties in defining a correct genotype–phenotype correlation and, therefore, a proper molecular diagnosis [6].

In light of this, the application of next-generation sequencing techniques, such as exome sequencing (ES), offers a powerful approach to identify and characterize new causative genes. Nevertheless, only the identification of additional families carrying mutations in the same genes provides definitive proof of their pathogenic role.

Here, we describe an Italian family affected by ADNSHL carrying a likely pathogenic variant in *NCOA3*, a gene recently described as a novel candidate for autosomal dominant progressive hearing loss in a Brazilian family [7].

## 2. Materials and Methods

The Italian family was recruited at the Institute for Maternal and Child Health—I.R.C.C.S. “Burlo Garofolo” (Trieste, Italy). All the patients provided written informed consent, and all research was conducted according to the ethical standard defined by the Helsinki Declaration.

The individuals of the family underwent a detailed clinical evaluation, including pure tone audiometric testing.

Subsequently, genomic DNA was extracted from peripheral whole blood samples using the QIAsymphony^®^ SP instrument with QIAsymphony^®^ Certal Kits (Qiagen, Venlo, The Netherlands), and DNA concentration was measured using a Nanodrop ND 1000 spectrophotometer (NanoDrop Technologies Inc., Wilmington, DE, USA).

A first round of genetic testing was carried out, including Sanger sequencing to analyze the entire coding region of the *GJB2* gene, multiplex PCR to search for *GJB6* deletions and Multiplex Ligation-dependent Probe Amplification (MLPA) analysis to identify deletion/duplication in the *STRC*, *CATSPER2* and *OTOA* genes with SALSA^®^ MLPA^®^ probe mixes P461-A1 DIS (MRC-Holland, Amsterdam, The Netherlands) according to the manufacturer’s protocol.

Finally, ES was carried out on an Illumina NextSeq 550 instrument (Illumina Inc., San Diego, CA, USA). According to the manufacturer’s instructions, genomic libraries were prepared using the Twist Human Core Exome + Human RefSeq Panel kit (Twist Bioscience, South San Francisco, CA, USA). The process leads to the creation of FASTQ files, the standard file format for DNA sequencing data. The FASTQ files were processed through a custom pipeline (Germline-Pipeline), developed by enGenome srl. This workflow allows the generation of a final VCF file containing information regarding variants such as single-nucleotide variants (SNVs), short insertion/deletions (INDELs) and exon-level copy number variations (CNVs). Thus, the VCF files were analyzed on enGenome Expert Variant Interpreter (eVai) software (evai.engenome.com). All the variants of interest were confirmed by Sanger sequencing. Finally, a linkage analysis was performed using Merlin software (version 1.1.2) [8], applying a dominant model with a disease allele frequency of 0.01% and penetrance 0,1.0,1.0. 

## 3. Results

A four-generation Italian family affected by ADNSHL was recruited at our institute (Figure 1).

Individuals underwent a careful clinical examination and dysmorphology assessment, which did not highlight any significant anomaly. Afterwards, a hearing evaluation by pure tone audiometry was performed on selected members of the family (II:4, III:1, III:3, III:6, III:8, III:10, III:11, III:13 and III:15), showing a bilateral, symmetric and moderate to severe hearing loss with a medium and high frequencies drop threshold profile (Figure 2A). The age of onset of the hearing impairment ranged from the fourth to the fifth decade of life. Moreover, patients complained of a worsening of their hearing performance over time, a feature objectively verified only in patient III:10 (Figure 2B). They also did not refer to any other comorbidities.

Fourteen family members underwent a first round of genetic tests, including analysis of mutations in the *GJB2* and *GJB6* genes, followed by MLPA analysis for the *STRC* gene, being one of the most common causes of hearing loss, second only to *GJB2*. All the tests resulted negative. ES was performed on all of the fourteen available family members. 

In particular, the analysis included two healthy individuals (II:3 and III:9) and members with severe and mild HL (II:4, II:5, III:1, III:3, III:5, III:7, III:8, III:10, III:13, III:15; III:6 and III:11) (Figure 2A).

The overall mean depth base coverage was 77.8X, while on average, 97% of the targeted region was covered at least 10-fold. Further details about the sequencing data are reported in Table 1. 

ES analysis led to the identification of a few possible causative variants. Considering the high number of tested individuals (fourteen subjects), after filtering (e.g., minor allele frequency lower than 0.01%, pathogenicity score, pattern of inheritance), the ES analysis led to the identification of only six missense variants in six different genes of particular interest due to their frequency, pathogenicity score given by the in silico tools and co-segregation with HL in most of the individuals of the family (see Table 2). Two of these missense variants are associated with phenotypes not present in any of our patients, such as autism spectrum disorder (*MBD2* and *SMC2*) [9,10]. In addition, one variant was located within a cancer driver candidate gene (*ALDH1B1* [11]) and another (*KCNH6*) resulted false positive after Sanger confirmation. Moreover, the co-segregation of these variants with the HL phenotype was not strictly consistent with the phenotype (see Table 2).

Finally, the last two variants (c.2909G>C in the *NCOA3* gene and c.9722T>A in the P*KHD1L1* gene) were located in genes encoding for proteins described as being involved in normal hearing function [7,12]. The variant located in *PKHD1L1* co-segregated with HL only in half of the patients, while the one within the *NCOA3* gene was in all the affected members of the family (Figure 3A), except for patient III:8 (Table 2). 

Moreover, linkage analysis for the *NCOA3* variant (rs765371222) showed an LOD score of 3.262. Thus, taking into account all these data and considering the high phenotype similarity between this family and the Brazilian family, the *NCOA3* variant was considered the likely causative one.

Many in silico predictors indicate the missense variant in the *NCOA3* gene (NM_181659.3, c.2909G>C, p.(Gly970Ala), rs765371222) as damaging; in particular, Polyphen2 showed a damage score of 0.82, MutationTaster a score of 0.999, PaPI algorithm a score of 0.908 and a DANN a score of 0.996 [13,14,15,16]. The application of the ACMG criteria for HL [17] allowed us to classify this variant as likely pathogenic (PM2, PP1_Strong). The search of this specific allele in an internal database of 892 Whole Genome Sequencing data highlighted its total absence in healthy individuals, supporting its possible pathogenic effect. The variant co-segregated with HL in all the affected individuals of the family, with the exception of patient III:8, who is probably affected by a different type of HL, likely due to noise exposure or other environmental factors (Figure 3A). 

Subsequently, other individuals of the fourth generation (IV:1, IV:2, IV:3, IV:4, IV:5, IV:6) were included in the study. Sanger sequencing was performed in the six subjects, confirming the presence of the variant in the individuals IV:1 and IV:5. They both underwent a hearing evaluation, which highlighted a mild hearing impairment in IV:1 and normal hearing in IV:5 (Figure 2A).

## 4. Discussion

The Nuclear Receptor Coactivator 3 (*NCOA3*) gene comprises 23 exons and encodes a protein of 1420 amino acids involved in the regulation of gene transcription [7]. Previous studies suggested a link between *NCOA3* variants and different pathologies, such as hypertriglyceridemia, obesity and dyslipidemia [18], or an association with the progression of post-traumatic osteoarthritis, bone mass, abnormal cartilage behavior and notch signaling pathway [7,19]. Recently, da Silva et al. described a clear correlation between the *NCOA3* gene and HL, showing the involvement of the protein in the development and physiology of the mouse auditory system and describing a family affected by ADNSHL carrying a missense variant in this gene [7]. In the present work, we reported the second family worldwide with a mutation within the *NCOA3* gene and a HL phenotype. Interestingly, both the Italian and the Brazilian missense variants (c.2909G>C (p.Gly970Ala), c.2810C>G (p.Ser937Cys)) are predicted as pathogenic by several in silico tools, and they are closely located (exon 15). Although this region corresponds to an unknown region domain (Figure 3B), the c.2909G>C variant causes the substitution of a highly conserved glycine with an alanine, both polar amino acids, but alanine contains an additional -CH_3_ group. The tetrahedral structure of the methyl group increases the steric hindrance of the amino acid residue, which might eventually influence the correct folding processes of the protein, compromising its stability or function (Figure 3C). Unfortunately, it has not been possible to generate a reliable in silico protein model due to the low characterization of the protein region carrying the variant. Nevertheless, its involvement has been further confirmed with linkage analysis. Moreover, the audiometric pattern observed in our family resulted to be consistent with the one of the Brazilian family; patients displayed a non-syndromic, progressive, sensorineural, bilateral, moderate to profound HL mainly involving the high and medium frequencies, reaching a severe to profound severity in some of the affected subjects (Figure 2A). The family is characterized by a progressive and late-onset HL, suggesting that the phenotype might be due to the slow but constant misfolded protein accumulation, leading to long-term damage of the ear structures. 

We investigated the co-segregation of the variant with HL in all the available family members, and it was found to be present in heterozygosis in all the affected individuals, except for patient III:8. Considering the high incidence of HL in the general population (~1–3:1000), it is not uncommon for large families to have at least one phenocopy (i.e., a phenotypic trait that resembles the trait expressed by a specific genotype, but in a subject who is not a carrier of that genotype). Given the high similarity of the studied family with the Brazilian one, it is likely that the HL phenotype of patient III:8 is due to other undermined causes. One possible explanation is her professional activity, since she is a factory worker constantly exposed to noise. Her audiometric profile is not exactly the one expected from noise-induced hearing loss; however, exceptions to this rule have already been described [20]. In addition, the absence of any other genetic cause leads us to consider her professional work as the most reasonable cause. 

The genetic co-segregation with HL was further extended to the fourth generation, allowing the early identification of HL in subject IV:1. Since the genetic result, the individual, who is 47 years old, underwent an audiometric exam confirming his clinical status. As regards individual IV:5, her hearing thresholds are in the range of normality, with a slightly worse performance at the middle and low frequencies. It is worth mentioning that she is only 32 years old and might develop HL in the following decades as her relatives did. Indeed, none of the affected individuals of the family reported having hearing problems before 40 years of age.

## 5. Conclusions

The identification of new genes and variants associated with monogenic forms of HL is extremely valuable since it allows for extending the knowledge of the molecular basis of this disease and providing patients with a recurrence risk estimation. When a new candidate gene is discovered, functional studies are essential to support genetic findings; however, an independent confirmation in other families (better if in a completely different ethnic group) is fundamental for definitely validating the result. In light of this, our results further confirm the involvement of *NCOA3* in the etiopathogenesis of ADNSHL, which should thus be considered a new gene for ADNSHL.

## Figures and Tables

**Figure 1 genes-12-01043-f001:**
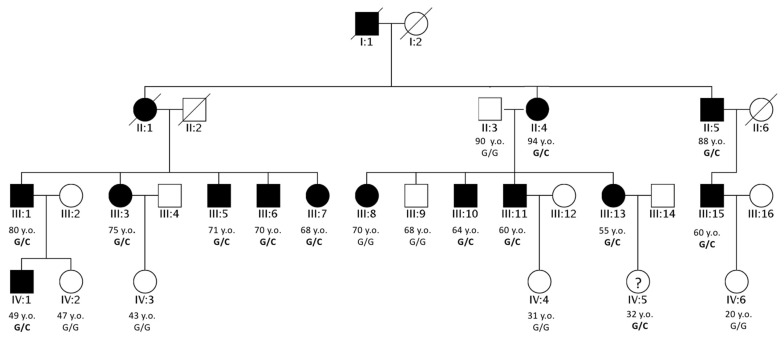
Pedigree of the Italian family. Pedigree of the four-generation family investigated in the present study. Filled symbols represent affected individuals. The age of each individual and the co-segregation of the *NCOA3* variant with HL (NM_181659.3, c.2909G>C, p.(Gly970Ala)) are reported. Their ages are indicated followed by the terms y.o., meaning years old. The labels used to report the subjects’ genotypes are G/G for homozygous wild type and G/C for the individuals carrying the variant herein discussed in heterozygosis. Individuals with Roman numeric labels were analyzed in this study.

**Figure 2 genes-12-01043-f002:**
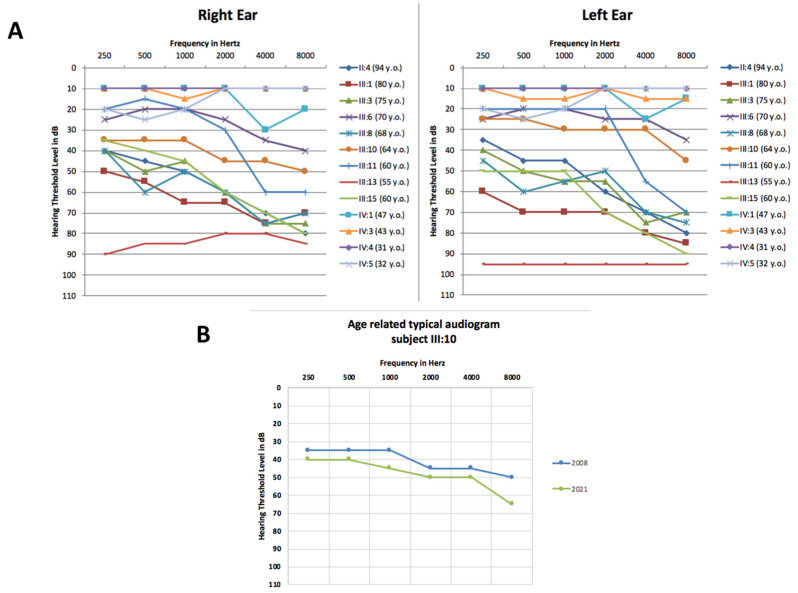
Audiograms of the Italian family. (**A**) Audiometric features, displayed as audiograms, were available for twelve members of the family. The thresholds of the right and left ears are shown for each patient. The audiometric patterns for patients IV:3 and IV:4 partially overlap. (**B**) Example of the progression of the hearing loss for patient III:10. In blue is displayed an audiogram performed in 2008, and in green, one of 2021 (Age-Related Typical Audiogram (ARTA)). Only the data for the best ear are reported.

**Figure 3 genes-12-01043-f003:**
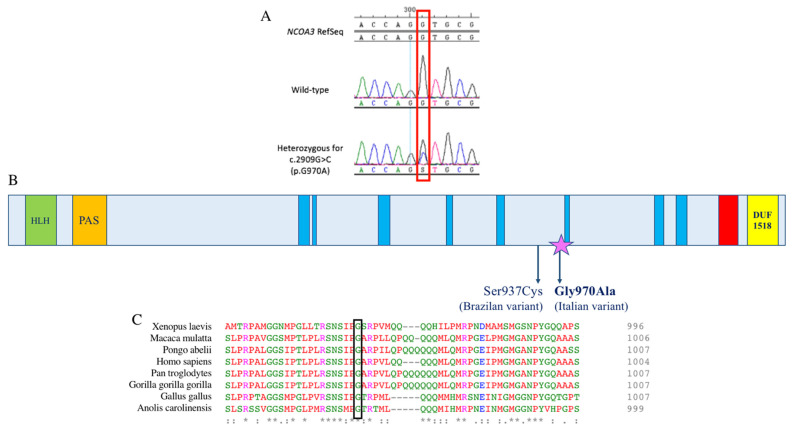
DNA chromatograms, schematic representation of the protein domains and protein alignment. (**A**) Chromatograms displaying part of the *NCOA3* sequence; on top, the wild-type sequence is shown, and on the bottom, that of an affected individual carrying the variant at the heterozygous state. The red box indicates the position of the variant. (**B**) Schematic representation of *NCOA3* protein domains and the localization of the two mutations associated with ADNSHL. The helix loop helix domain is displayed as a green box; in orange, the PAS motif; in blue, low-complexity regions; in red, the coiled coil region; and in yellow, the SMART DUF1518 domain, commonly found in receptor co-activating proteins. Both the variants detected in the *NCOA3* gene and associated with ADNSHL are located in the same undefined region of the protein; the one found in the Italian family is indicated with a purple star. (**C**) Protein alignment showing conservation of residue G970 across species.

**Table 1 genes-12-01043-t001:** Next-generation sequencing data. Data relative to ES for each patient. The bold character highlights to which patient the data presented refers to.

	Mean Target Coverage (Mean Base Coverage Depth)	Coverage Uniformity (Percentage of Bases with a Coverage Depth > 0.2 × Mean_Coverage)	% Target Bases 10X (Percentage of Target Bases with Coverage above 10X)
**II:3**	78.469824	97.5476	97.4332
**II:4**	77.629201	95.2360	98.0183
**II:5**	79.013458	96.1209	97.0431
**III:1**	77.012397	98.0127	96.0126
**III:3**	78.723768	97.5421	98.3218
**III:5**	77.629071	96.9301	97.9854
**III:6**	77.204720	95.9183	97.4510
**III:7**	75.927492	96.5333	96.9981
**III:8**	76.638206	98.1426	95.4832
**III:9**	78.656329	95.6721	96.9453
**III:10**	76.739260	98.2310	97.0034
**III:11**	78.739279	96.9231	95.8741
**III:13**	78.317429	97.8342	96.7632
**III:15**	79.125820	96.7631	97.1632

**Table 2 genes-12-01043-t002:** Genotype–phenotype associations. Co-segregation of the six variants with HL in the fourteen members of the family for whom ES was performed. Each patient’s age and gender are indicated. The underlined genotypes are not consistent with the phenotypes of the patients, while the ones reported in bold highlight the segregation of the most interesting variant (NM_181659.3, c.2909G>C, p.(Gly970Ala), rs765371222) in the affected individuals.

Subject	Affection	Variants					
		***ALDH1B1*** NM_000692.5	***SMC2*** NM_001042550.2	***NCO3*** NM_181659.3	***MBD2*** NM_015832.6	***KCNH6*** NM_030779.4	***PKHD1L1*** NM_177531.4
		c.G1087A	c.A1514G	c.G2909C	c.G707T	c.G263T	c.T9722A
II-3 (male, 92 y.o)	healthy	G/G	A/A	G/G	G/G	false positive	T/T
II-4 (female, 94 y.o.)	**affected**	G/A	A/G	**G/C**	G/T	false positive	T/A
II-5 (male, 88 y.o.)	**affected**	G/A	A/G	**G/C**	G/T	false positive	T/A
III-1 (male, 80 y.o.)	**affected**	G/A	A/G	**G/C**	G/T	false positive	T/T
III-3 (female, 75 y.o.)	**affected**	G/A	A/G	**G/C**	G/T	false positive	T/A
III-5 (male, 71 y.o.)	**affected**	G/G	A/A	**G/C**	G/G	false positive	T/T
III-6 (male, 70 y.o.)	**affected**	G/G	A/G	**G/C**	G/G	false positive	T/T
III-7 (female, 68 y.o.)	**affected**	G/A	A/G	**G/C**	G/G	false positive	T/T
III-8 (female, 70 y.o.)	**affected**	G/G	A/G	G/G	G/G	false positive	T/T
III-9 (male, 68 y.o.)	healthy	G/G	A/G	G/G	G/T	false positive	T/T
III-10 (male, 64 y.o.)	**affected**	G/A	A/G	**G/C**	G/T	false positive	T/A
III-11 (male, 60 y.o.)	**affected**	G/G	A/G	**G/C**	G/G	false positive	T/T
III-13 (female, 56 y.o.)	**affected**	G/A	A/G	**G/C**	G/T	false positive	T/A
III-15 (male, 60 y.o.)	**affected**	G/A	A/G	**G/C**	G/T	false positive	T/A

## Data Availability

The data presented in this study are available on request from the corresponding author. The data are not publicly available due to privacy restrictions.

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
