# Peer review of "Non-Syndromic Autosomal Dominant Hearing Loss: The First Italian Family Carrying a Mutation in the NCOA3 Gene"

_genes, 2021, doi:10.3390/genes12071043_

Round 1

Reviewer 1 Report

In this manuscript, the authors describe the identification of a variant in NCOA3 gene possibly associated with Non-Syndromic autosomal dominant hearing loss in an Italian family.

I have several considerations that I show below:

1-Material and methods line 76 to 85. NGS sequencing data need to be improved; for example depth, coverage, etc should be shown.

2- Line 100 Author affirm “…was slowly progressive over time “ They should be shown more data for example an ARTA analysis.

3-Patients hearing threshold level should be shown as a graph. Moreover, the age of patients should be shown

4- The variant identified should be classified according to the ACMG criteria. According to my data;  p.(Ser937Cys) ( the Brazilian mutation) should be classified as likely benign (BS1 BP1) and p.(Gly970Ala)  as VUS (PM2 BP1).

5- Please use the protein  nomenclature as follows: p.(Gly970Ala)

6- Protein alignment should show other organisms non-mammals for example Xenopus, Drosophila, etc to know the conservation. Moreover, the species shown are evolutionarily too close.

7- line 168 “ are predicted as pathogenic by all in silico tools used” I have analyzed both variants and in the case of p.(Ser937Cys) is classified as pathogenic by 4 predictors and benign by 14. For p.(Gly 970Ala) 5 pathogenic and 14 benign. Please verify this point.

8 Line 186 -187. The authors affirm that the audiometric profile of patient III:8 could be explained by high exposure to noise due to work activity) but I have seen that she does not exhibit the typical audiometric profile cause by noise exposure.

In conclusion, I have doubts about the pathogenicity of the variant described by the authors. Maybe functional assays should be performed to clarify the role of this variant.

Reviewer 2 Report

The authors of „Non-syndromic autosomal dominant hearing loss: the first Italian family carrying a mutation in NCOA3 gene “ delivered a valuable contribution to the field of hearing disorders caused by genetic mutations. In addition, the paper adds new information on this rare mutation.

I only have a few minor comments:

Please use small caps when listing the names of amino acids (lines 170 – 171).

  1. In Table 2, please add the age and gender of the patients. Although figure 1 already presents the information on gender, one has to jump from figure to table.
  2. I suggest presenting the data from Table 2 in the form of an audiogram. Thus, one can better appreciate the shape of the slope.
  3. Please add a few sentences to discuss the implication of a correlation between age and the HL phenotype in patients affected by the mutation.
  4. Apart from the dysmorphology assessment – was the medical history of the patients also collected? Were there any comorbidities observed (e.g., orthopedic conditions)?

Reviewer 3 Report

This manuscript identified a second mutation in the NCOA3 gene, in a second family, worldwide, with HL caused by a variant in this gene. Independent confirmation of the same gene in more than one family could serve as significant validation of the gene as involved in HL. However, in this reported family the segregation was not complete, as one hearing impaired did not carry the variant and another unaffected member did carry it. These discrepancies could be explained if further information is obtained regarding the misfit individuals, as detailed below.

The manuscript is formulated very clearly and comprehensively and adds important supportive data for the validation of a recent reported new ADNSHL.

P.3, Fig.1: Please indicate age (year of birth) and age of onset for all individuals analyzed.

P.3, Table 1: To my opinion, the table should be replaced by audiograms, as in the Brazilian paper. Moreover, as the difference between the ears is not big, and the HL is considered bilateral symmetric, it would be enough to present, for each individual, just one ear (the better one) or, alternatively, the average of both ears.

P.5, L.152: Please indicate the ages of IV:1 and IV:5.

P.5, L. 154-155: an audiogram is essential for IV:5.

Discrepancy: on P.3, L.99 is written: “onset of the hearing impairment ranged from the fourth to the fifth decade of life”, however, on P.6, L.178-179 is written: “a variable age of onset, ranging from childhood to adulthood.” – Please correct.

P.6, L.186-187: Can you please find supportive evidence for the cause of HL in individual III:8? e.g., occupation (to support noise exposure), usage of ototoxic drugs, etc.

Round 2

Reviewer 1 Report

The manuscript has been improved.

On the other hand, I think that there is not enough evidence to assume that  NCOA3 is a gene associated with Hearing loss, then the phrase " Our findings (i.e., the identification of the second family reported globally with HL caused by a variant in NCOA3) further confirm the involvement of NCOA3 in the etiopathogenesis of ADNSHL, which should thus be considered as a new gene for autosomal dominant non-syndromic hearing loss" should be changed, mainly the word confirm.
